# An Effective Design Formula for Single-Layer Printed Spiral Coils with the Maximum Quality Factor (Q-Factor) in the Megahertz Frequency Range

**DOI:** 10.3390/s22207761

**Published:** 2022-10-13

**Authors:** Young-Jin Park, Ji-Eun Kim, Su-Hyeong Lee, Kyung-Hwan Cho

**Affiliations:** Korea Electrotechnology Research Institute, An-San 15588, Korea

**Keywords:** magnetic resonant wireless power transfer, quality factor (Q-factor), printed spiral coil, ohmic losses, volume filament model

## Abstract

This paper presents a design formula for a printed spiral coil to ensure the maximum quality factor (Q-factor). The formula is composed of a pattern’s metal thickness, single pattern width, total pattern width, and turn number, and is effective in the megahertz (MHz) frequency range. During the formula’s design, the resistance, self-inductance, and Q-factor are calculated according to the ratio of each pattern’s width and total pattern width and the turn number for different metal thicknesses, frequencies, and total pattern widths using a volume filament model (VFM). With a given turn number and metal thickness, the optimal ratio of individual and total pattern widths can be determined to ensure the maximum Q-factor. To verify the formula, some optimal coils were fabricated, and the calculations and measurements were shown to have good agreement. Furthermore, the optimized coils were shown to have higher coupling efficiency than the coils without optimal dimensions.

## 1. Introduction

An effective coil with an improved quality factor (Q-factor) is essential in magnetic resonance wireless power transfer (MR-WPT). The Q-factor of a coil (=2*πfL/R*) is calculated from its frequency (*f*), self-inductance (*L*), and losses (*R*). Therefore, it is important to design a coil with higher self-inductance and lower losses in the megahertz (MHz) frequency range.

Coil losses mainly consist of skin resistance and proximity resistance in the MHz frequency range. The skin effect is caused by an eddy current in a metal wire, which then increases skin resistance [1] at higher frequencies. The proximity effect also results from the current induced in the metal wire via a magnetic field generated by neighboring metal wires. This effect reduces the effective area of the current flow, leading to a very quick increase in ohmic resistance, especially when many wires are bundled tightly. Therefore, a metal wire is preferable to a Litz wire in the MHz frequency range.

In the MHz frequency range, a printed spiral coil is very often used for the MR-WPT of biomedical sensors, Internet-of-Things devices, and consumer devices owing to its easy fabrication and compactness [2,3,4,5,6,7,8]. In [9], a compact flexible spiral coil was designed to use wireless power transfer (WPT) and near-field communication (NFC) at 13.56 MHz. In [10], a viaed compact broadband planar double spiral coil was explored for a robust WPT system at 50 MHz. In [11], the design topology of a printed spiral coil with unequal width and pitch for a high Q-factor was presented and designed at 6.78 MHz. In [12,13,14], printed rectangular spiral coils were reported. The previous references show that the design of a printed spiral coil with lower ohmic resistance through a reduction in both skin and proximity resistance is important.

Several resistance calculation and optimization methods for printed spiral coils have been researched. Lope et al. [15] reported an analytical solution, but it was only effective at frequencies below about 1 MHz. Nguyen and Blanchette [16] calculated the total resistance by determining a skin resistance and proximity factor, but their method used less than about 1 MHz, as skin resistance is effective only at less than 1 MHz. Kim and Park [14] reported a calculation method for the design of an effective planar rectangular coil with a lower resistance using a numerical volume filament model (VFM). However, they analyzed a printed rectangular coil and did not consider metal thickness. Jow and Ghovanloo [17] used an approximate calculation method for resistance and self-inductance in the case of printed rectangular spiral coils in the MHz frequency range, but did not show the analysis results on ohmic losses and self-inductance according to pattern metal thickness. In all of these previous works, no optimal practical design method of a printed spiral coil for effective WPT in the MHz frequency range was presented that considered the resistance and self-inductance of a printed spiral coil.

The current paper presents a formula for the optimal design of a printed spiral coil to ensure the maximum Q-factor. The Q-factor of a coil’s WPT is directly related to its efficiency [18]. To derive the formula, the inductance and ohmic resistance of printed spiral coils are calculated for the parameters turn number, pattern width, metal thickness, and total pattern width using a VFM at both 2 MHz and 6.78 MHz. The results of this calculation reveal an optimal formula to ensure the maximum Q-factor of the coil parameters.

Regarding the main contributions of the paper, it is newly shown that the optimal relationship of pattern width and total pattern width for maximum Q-factor can be determined by the pattern thickness and number of turns, while it is little changed by the operating frequency being in the MHz frequency range. Moreover, an effective design formula including the turn number, metal thickness, pattern width, and pattern gap for the design of printed spiral coils with the maximum Q-factor is newly provided for the case of printed spiral coils. The effects of metal thickness on Q-factor are likewise presented.

This paper is organized as follows. Section 2 explains the resistance and self-inductance calculations for printed spiral coils using VFM. Then, Section 3 reports the calculation and analysis results of this study. In particular, Q-factors according to *w*0/*W* and the turn number are displayed for different frequencies, metal thicknesses, and total pattern widths. From the calculated Q-factor, an approximate formula to ensure the maximum Q-factor is derived. A precise formula is then derived based on the former calculation. Finally, Section 4 verifies the proposed formula by comparing measurements and calculations.

## 2. Calculating the Resistance and Self-Inductance of a Printed Spiral Coil Using a VFM

### 2.1. Discretization of a Printed Spiral Coil for a VFM

In this section, a VFM is used to calculate the resistance and self-inductance of a printed spiral coil.

Figure 1a,b shows a printed spiral coil and a concentric multi-loop coil that is equivalent to the spiral coil for the purposes of analysis and calculation. Figure 1c displays a cross-section of the printed spiral coil. In the figure, *w*0 and *t* represent a pattern’s width and thickness, respectively, and *g* represents the gap between neighboring patterns. *R_out_* and *N* denote the radius of the outermost loop and the turn number of the coil, respectively. *W* indicates the total pattern width.

To apply the VFM to a printed spiral coil, the equivalent coil in Figure 1b is discretized, as shown in Figure 2. Figure 2a depicts a rectangular cross-section of the *k*^th^ and *m*^th^ loops with square discretization in a printed concentric coil of *N* loops. *I*_1_, *I_k_*, *I_m_*, and *I_N_* represent the currents in each loop. For the VFM, the *k*^th^ and *m*^th^ loops are divided into *n* circular filament loops of square cross-sections with *d* width and height. For the calculation, *d* is set to 0.2*δ*, where *δ* is the skin depth [19]. Figure 2b shows two circular filament loops, *c_kf_* and *c_mg_*. Here, *c_kf_* represents the *f*^th^ filament loop of the *k*^th^ loop and *c_mg_* represents the *g*^th^ filament loop of the *m*^th^ loop. The radii of *c_kf_* and *c_mg_* are denoted by *s_kf_* and *s_mg_*, respectively. *D_kf_mg_* represents the distance between *c_kf_* and *c_mg_*. Then, as shown in Figure 2b, two filament loops of *c_kf_* and *c_mg_* are represented by the resistances of *r_kf_* and *r_mg_*, self-inductances of *L_kf_* and *L_mg_*, and the mutual inductance of *M_kf_mg_*.

### 2.2. The Equivalent Circuit and Formulation for Calculation

Figure 3 shows the equivalent circuit of the concentric multi-loop coil, which is discretized into circular filament loops. The *f*^th^ filament loop of *c_kf_* is represented by the self-inductance *L_kf_* and the resistance r*_kf_*. *V_k_* and *I_k_* respectively indicate the voltage and current at the *k*^th^ loop, while *v_kf_* and *i_kf_* indicate the voltage and current at *c_kf_*. *M_kf_mg_* represents the mutual inductance between *c_kf_* and *c_mg_*.

Applying Kirchhoff’s law provides the following matrix [14]:(1)$$\left[\begin{array}{c} V_{1}\\ \vdots \\ \begin{array}{c} V_{k}\\ \vdots \\ V_{m} \end{array}\\ \vdots \\ V_{N} \end{array}\right]=\left[\begin{array}{ccccccc} R_{1}+j\omega M_{11} & \cdots & j\omega M_{1k} & \cdots & j\omega M_{1m} & \cdots & j\omega M_{1N}\\ \vdots & \ddots & \vdots & & \vdots & & \vdots \\ j\omega {M^{T}}_{1k} & \cdots & R_{k}+j\omega M_{\mathrm{kk}} & \cdots & j\omega M_{km} & \cdots & j\omega M_{kN}\\ \vdots & & \vdots & \ddots & \vdots & & \vdots \\ j\omega {M^{T}}_{1m} & \cdots & j\omega {M^{T}}_{km} & \cdots & R_{m}+j\omega M_{\mathrm{mm}} & \cdots & j\omega M_{mN}\\ \vdots & & \vdots & & \vdots & \ddots & \vdots \\ j\omega {M^{T}}_{1N} & \cdots & j\omega {M^{T}}_{kN} & \cdots & j\omega {M^{T}}_{mN} & \cdots & R_{N}+j\omega M_{NN} \end{array}\right]\left[\begin{array}{c} I_{1}\\ \vdots \\ \begin{array}{c} I_{k}\\ \vdots \\ I_{m} \end{array}\\ \vdots \\ I_{N} \end{array}\right],$$
where
$$I_{k}=[i_{k1}\cdot \cdot \cdot \;i_{kf}\cdot \cdot \cdot i_{mg}\cdot \cdot \cdot \;i_{kn}],\;\;V_{k}=[v_{k1}\;\cdot \cdot \cdot \;v_{kf}\cdot \cdot \cdot v_{mg}\cdot \cdot \cdot v_{kn}]$$

In (1), ω represents the angular frequency and [∙]^T^ represents the transverse matrix of [∙]. ***R****_k_* represents the diagonal matrix of *r_kf_*. ***M****_km_* is a matrix of *M_kf_mg_* (*k*, *m* = 1, …, *N* and *f*, *g* = 1, …, *n*). When *k* = *m* and *f* = *g*, *M_kf_mg_* represents the self-inductance of *c_kf_*, *L_kf_*.

As the current distribution of each filament loop can be assumed to be uniform, the resistance of each filament loop, *r_kf_*, is expressed as the total length (2*πs_kf_*) divided by the cross-section area and conductivity. That is,
(2)$$r_{kf}=\frac{2\pi s_{kf}}{\sigma d^{2}}$$
where *σ* is the copper conductivity (*σ* = 5.96 × 10^7^ S/m).

The inductance of *c_kf_*, *L_kf_*, is calculated as follows [20]:(3)$$L_{kf}=\mu_{0}s_{kf}\left[\ln\frac{8s_{kf}}{\left(d/2\right)}-1.75\right]$$
where *μ*_0_ is the permeability in vacuum (*μ*_0_ = 4π × 10^−7^).

Given that the two filament loops of *s_kf_* and *s_mg_* are perfectly aligned, the mutual inductance between *c_kf_* and *c_m_*_g_, *M_kf_mg_* is calculated as follows [20]:(4)$$M_{kf\_mg}=\mu_{0}\sqrt{s_{kf}s_{mg}}\left[\left(\frac{2}{\kappa }-2\right)K(\kappa )-\frac{2}{\kappa }E(\kappa )\right]$$
where
(5)$$\kappa =\sqrt{\frac{4s_{kf}s_{mg}}{(s_{kf}^{2}+s_{mg}^{2})+D_{kf\_mg}^{2}}}.$$

*K*(*κ*) and *E*(*κ*) are complete elliptic integrals of the first and second kind, respectively [21].

The net currents of the loops are identical, as all loops of a coil will be connected in series practically. For the simple calculation of resistance and self-inductance in (7), it is also assumed that the current of all loops is the same as 1 A. That is,
(6)$$I_{1}=I_{2}=\cdot \cdot \cdot =I_{k}=\cdot \cdot \cdot =\;I_{N}=\;1\;A$$
where
$$I_{k}=\sum_{f=1}^{n}i_{kf}$$

Combining (1) and (6) can derive the voltage across the *k*^th^ loop, *V_k_*. If a current of 1 A flows through a coil, then the total voltage across the coil will be equal to the coil’s impedance. Thus, the resistance and self-inductance of the coil, *R* and *L*, are as follows:(7)R=Re(∑k=1NVk),  L=Im(∑k=1NVk)/ω .
where Re(∙) and Im(∙) are the real and imaginary parts of (∙), respectively.

### 2.3. Calculation Results and Analysis

A coil’s Q-factor is defined as *ωL*/*R*, where *L* and *R* indicate the coil’s self-inductance and ohmic resistance (or ohmic losses), respectively. As the Q-factor is related to efficiency in a WPT system [18], designing a coil with a higher Q-factor is very important.

Using the calculation formulation from (1) to (7) explained in the previous section, the resistance and self-inductance of a printed spiral coil according to *w*0/*W* and *N* are calculated for frequencies of 2 MHz and 6.78 MHz; metal thicknesses of *t* = 1 oz, 2 oz, 4 oz, and 6 oz; total pattern widths (*W*) of 10 mm and 30 mm; and *R_out_* of 100 mm. The Q-factors for these cases were obtained using the calculated self-inductances and resistances.

First, the proposed calculation method was verified through simulation using ANSYS Maxwell. In Figure 4, the calculation and simulation are compared for *W* = 30 mm, *t* = 2 oz, *f* = 6.78 MHz, and *N* = 10. The maximum differences between the two methods for resistance and self-inductance are 1.27% and 0.0062%, respectively. Specifically, the calculations of resistance and self-inductance were shown to be consistent with the calculations of simulations, according to *w*0/*W*. In addition, note that the lowest resistance existed near *w*0/*W* = 0.075. This was because when *w*0/*W* < 0.075, skin resistance (*R_skin_*) increased and proximity resistance (*R_prox_*) increased rapidly as the gap between two neighboring patterns (*g*) was reduced when *w*0/*W* > 0.075. To explain this, the proximity factor (*G_p_*) was obtained using the average resistance (*R_avg_*) and skin resistance (*R_skin_*) [22], as follows:(8)$$R_{avg}=R_{skin}+R_{prox}=R_{skin}(1+R_{prox}/R_{skin})=R_{skin}(1+G_{p})\;(\Omega /m)$$
where
(9)$$G_{p}=R_{avg}/R_{skin}-1.$$

*R_avg_* indicates the average ohmic resistance of a pattern loop per unit length obtained at approximately *W* = 30 mm, as follows:(10)$$R_{avg}=\frac{1}{N}\frac{R}{2\pi (R_{out}-15)/1000}(\Omega /m)$$
where *R* is the total ohmic resistance calculated using (7) and *N* = 10. In Figure 5, normalized *R_skin_* (*R_norm_*) is the skin resistance normalized for *w*0/*W* = 0.005 and *G_p_* is the proximity factor. As shown in Figure 5, the skin resistance increased for *w*0/*W* < 0.075, while for *w*0/*W* > 0.075, *G_p_* increased rapidly while the skin resistance decreased gradually. Therefore, the lowest resistance occurred near *w*0/*W =* 0.075.

In Figure 6, the calculated Q-factors are displayed according to 0.005 ≤ *w*0/*W* ≤ 0.25 and 2 ≤ *N* ≤ 20 at frequencies of 2 MHz and 6.78 MHz; metal thicknesses of 2 oz, 4 oz, and 6 oz; and total pattern widths of 10 mm and 30 mm. Additionally, for explanatory purposes, the resistances, self-inductances, and Q-factors are indicated for *N* = 5, 10, and 15. The dark blue color in the figure indicates that a coil cannot be produced.

As Figure 6 shows, an optimal *w*0/*W* for the maximum Q-factor can be found for each turn number in all cases. Note as well that the optimal *w*0/*W* was similar for each turn number in all cases, although the Q-factor increased with a thicker pattern, higher frequency, and wider total pattern for the same turn number. Furthermore, the optimal *w*0/*W* for the maximum Q-factors with the same frequency and pattern width was found to change slightly according to *t.*

Note that self-inductances were similar with the same turn number, total pattern width, and pattern width—regardless of frequency and thickness—while the ohmic resistances changed according to turn number, total pattern width, pattern width, frequency, and metal thickness.

According to the turn numbers, the maximum Q-factor changed little in all cases. This finding indicates that a turn number can be derived for self-inductance, which is necessary for a WPT system, and an optimal *w*0/*W* can then be obtained for the maximum Q-factor and maximum coupling efficiency.

As Figure 6 indicates, as a pattern thickened, the maximum Q-factor can be improved at the same frequency and *W*, as the self-inductance remained nearly the same while the ohmic resistance decreased. Practically, however, fabrication costs will increase for patterns thicker than 4 oz. The maximum Q-factor can also be improved with a higher frequency, because self-inductance changed very little while ohmic resistance grew slightly.

Figure 6a also shows that the ohmic resistance certainly decreased when the metal thickness changed from 2 oz to 4 oz for the same turn number and optimal *w*0/*W*, while it decreased slightly when the metal thickness changed from 4 oz to 6 oz. Figure 6c shows that the ohmic resistance decreased slightly when the metal thickness changed from 2 oz to 4 oz for the same turn number and optimal *w*0/*W*, while it rarely decreased when the metal thickness changed from 4 oz to 6 oz. Therefore, an improvement in ohmic resistance could not be obtained effectively when the metal thickness exceeded 4 oz at 2 MHz or 2 oz at 6.78 MHz. This finding indicates that a metal thickness of 4 oz at 2 MHz or 2 oz at 6.78 MHz could be used for optimal coil design.

Comparisons between Figure 6a,c or Figure 6b,d reveal that ohmic resistance increased at a higher frequency while self-inductance remained nearly the same for the same total pattern width, turn number, and *w*0/*W*. The reason for this finding is that skin resistance decreased at a lower frequency because the skin depth was greater.

### 2.4. Deriving a Formula to Ensure the Maximum Q-Factor and Comparing Formulas and Calculations

As Figure 6 shows, the optimal *w*0/*W* for a given turn number was similar across all cases. Figure 7 indicates the optimal *w*0/*W* for the maximum Q-factor according to a given turn number at frequencies of 2 MHz and 6.78 MHz and pattern widths (*W*) of 10 mm and 30 mm, based on the calculation results presented in Figure 6. For metal thicknesses (*t*) of 2 oz, 4 oz, and 6 oz, the following formula was derived using curve fitting:(11)$$\frac{w0}{W}=-0.00156\times t(\mathrm{oz})+0.754185\times N^{-0.9263}-0.01122,\;N\geq 3.$$

Using (11), an optimal relationship of *w*0/*W* for the maximum Q-factor can be determined with a given metal thickness and turn number. For the total pattern width, each pattern’s width and the gap between neighboring patterns were also determined. Note that the formula includes the effect of metal thickness. In Figure 7, the calculation results of the proposed formula are compared for *t* = 2 oz, 4 oz, and 6 oz. Note that, with thicker patterns, the optimal *w*0/*W* decreased slightly for *N* ≥ 5. The formula’s results are shown consistent with the calculation results in Figure 6.

Note that the formula is effective in the MHz frequency range with total pattern widths less than about *R_out_*/2 in cases where the proximity effect by a reverse current can be ignored [19].

## 3. Fabrication and Comparisons between Measurements and Calculations

To verify the paper’s calculations, some optimized and non-optimized coils were fabricated. Figure 8 presents these fabricated coils. Table 1 displays the specifications of the fabricated coils, such as *t*, *W*, *w*0/*W*, and *N*. Coils 1, 2, 3, and 4 were produced using flexible PCB, while coils 5 and 6 were produced using FR4. Coils 1 and 2 were optimized, while coils 3 and 4 were not. The Agilent E5072A network analyzer was used to obtain measurements.

In Table 1, the Q-factors measured at 2 MHz are compared with corresponding calculations. First, the self-inductance measurements were consistent with the corresponding calculations, while the ohmic resistance measurements were slightly higher than the corresponding calculations. Therefore, the measured Q-factors were slightly lower than the calculated Q-factors. The reason for this difference was that the fabrication errors and equivalent series resistances (ESRs) for lumped capacitors at a resonance of 2 MHz can slightly increase the total ohmic resistance. That is, in fabrication, coil patterns were not etched precisely and the coil dimension may have been slightly changed. Small dielectric loss was also added. Additionally, lumped capacitors at a resonance of 2 MHz were used, and the ESR of the capacitors was added. To reduce the ESR, the resonant capacitors were loaded in parallel. Coils 3 and 4 were not optimized for maximum Q-factors. Compared with coils 1 and 2, coils 3 and 4 had lower Q-factors, increased ohmic resistance, and similar self-inductance.

For comparison between coils 2 and coil 7, the Q-factor for the metal pattern of 4 oz was higher than that for 2 oz and the ohmic resistance was lower, while the self-inductances of the two coils were similar. From the measurement of coils 1 and 5, the optimal *w*0/*W* for the same turn number was the same, and for coils 2, 6, and 7, the optimal *w*0/*W* for the same turn number was similar, but with different W and metal thicknesses.

The maximum coupling efficiencies, *η*, were compared between the two coils with the highest Q-factors. Compared with the optimal coils, these efficiencies were calculated as follows:(12)$$\eta =\frac{FoM^{2}}{{\left(\sqrt{1+FoM^{2}}+1\right)}^{2}}=\frac{\sqrt{1+FoM^{2}}-1}{\sqrt{1+FoM^{2}}+1}\;\mathrm{with}\;FoM=\frac{\omega M_{12}}{\sqrt{R_{1}\cdot R_{2}}}.$$
where *R*_1_ and *R*_2_ indicate the resistances of the Tx and Rx coils, respectively; *M*_12_ represents the mutual inductance between Tx and Rx coils; and *FoM* indicates the figure of merit of the MR-WPT system.

Table 2 presents the maximum coupling efficiency according to the distance between the Tx and Rx coils calculated with Equation (12). Mutual inductances were measured using the Agilent E5072A network analyzer, and measured resistances were used to determine the *FoM*. In Table 2, the maximum coupling efficiency between two coils is displayed at 15 cm, 30 cm, and 45 cm. Note that, as the turn number increased, the mutual inductance for two coils of 10 turns increased, but the efficiency and *FoM* remained similar compared with Case 1 of coils 1 and 2. The reason for this finding was that ohmic resistance also increased as the turn number increased. Therefore, an optimal printed spiral coil to target a specific self-inductance or turn number could be designed for a given total pattern width and metal thickness, and the coupling efficiency will be similar across such optimized coils, regardless of their turn number.

Finally, this work is compared with the previous works in Table 3. As shown, an optimal design formula for the maximum Q-factor is provided in this paper. The formula can be applied for various metal thicknesses in the MHz frequency range.

## 4. Conclusions

In this paper, an optimal design formula to ensure the maximum Q-factor of a printed spiral coil is provided for MR-WPT applications in the MHz frequency range. Accordingly, the ohmic resistance, self-inductance, and Q-factors of the printed spiral coils were calculated according to 0.005 ≤ *w*0/*W* ≤ 0.25 and 2 ≤ *N* ≤ 20 at frequencies of 2 MHz and 6.78 MHz, metal thicknesses of 2 oz, 4 oz, and 6 oz, and total pattern widths of 10 mm and 30 mm. An optimal *w*0/*W* to ensure the maximum Q-factor was found to be determinable for each turn number. Moreover, the optimal *w*0/*W* was found to be similar for each turn number in all cases, although the Q-factor was found to increase with thicker patterns, higher frequencies, and wider total pattern widths for the same turn number. With thicker patterns, the optimal *w*0/*W* was found to decrease slightly for *N* ≥ 5, regardless of frequencies. Finally, an optimal design formula was provided for a single-layer printed spiral coil to ensure its maximum Q-factor in the MHz frequency range. This formula can also be used to design an optimal printed spiral coil when a specific self-inductance is required.

## Figures and Tables

**Figure 1 sensors-22-07761-f001:**
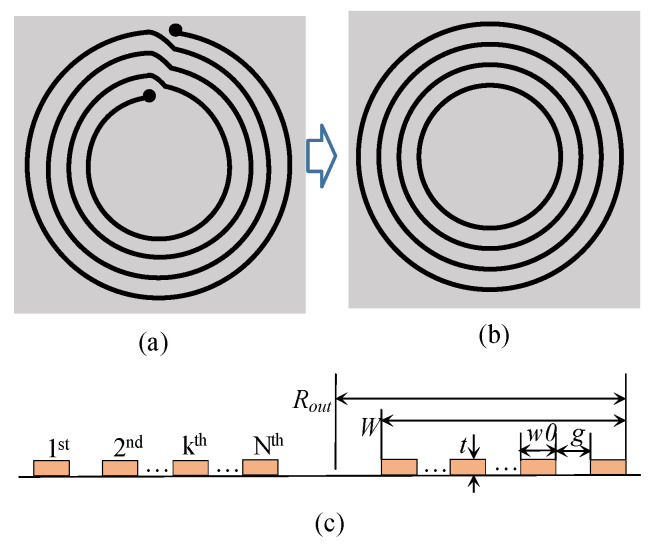
Illustrations of (**a**) a printed spiral coil, (**b**) a concentric multi-loops coil equivalent to the spiral coil, and (**c**) their cross-sections.

**Figure 2 sensors-22-07761-f002:**
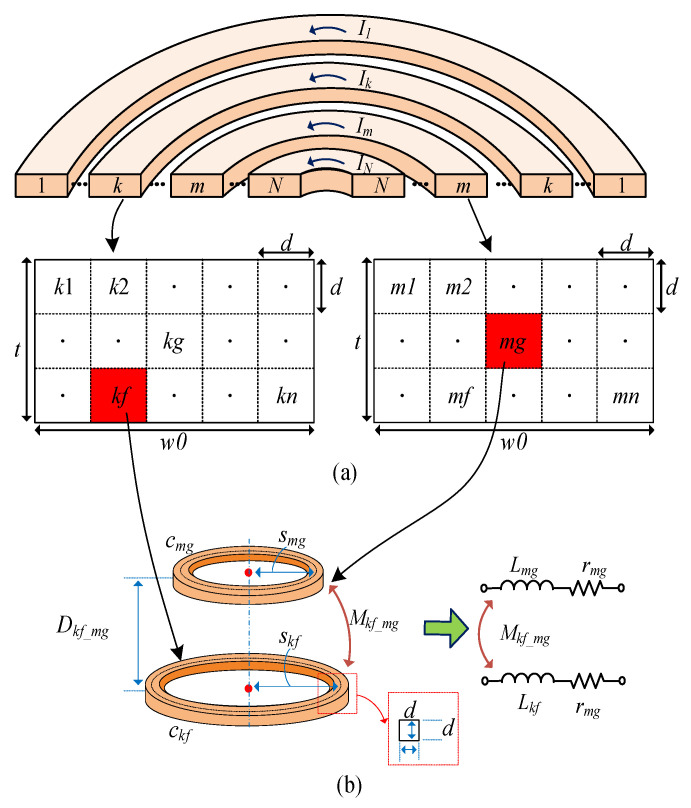
(**a**) Printed concentric circular coils of *N* loops and cross-section of the *k*^th^ and *m*^th^ loops with discretization in the square. (**b**) The *f*^th^ and *g*^th^ circular filament loops of the *k*^th^ and *m*^th^ circular loops and their equivalent model.

**Figure 3 sensors-22-07761-f003:**
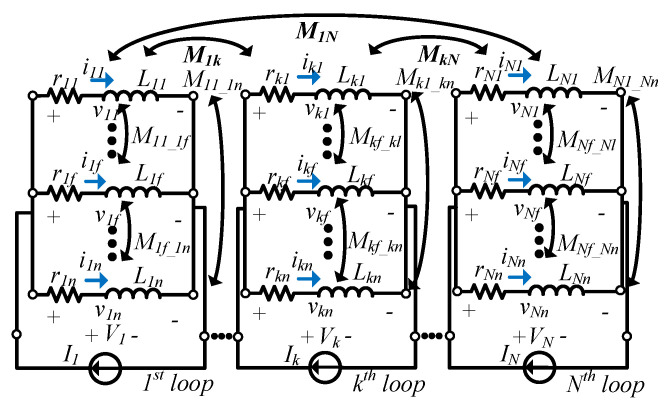
An equivalent circuit of a printed spiral coil with *N* turns.

**Figure 4 sensors-22-07761-f004:**
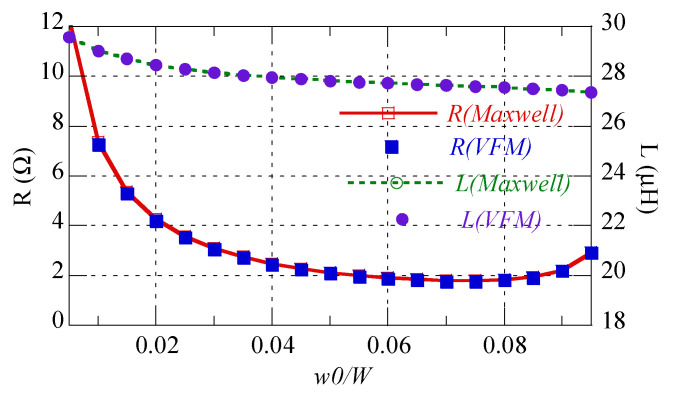
Comparison between the calculations and simulations of ohmic resistance and self-inductance for *W* = 30 mm, *t* = 2 oz, *f* = 6.78 MHz, and *N* = 10.

**Figure 5 sensors-22-07761-f005:**
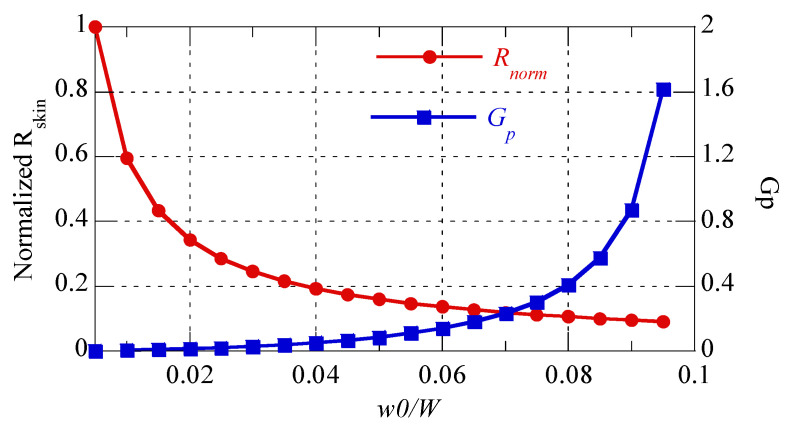
Normalized skin resistance (*R_norm_*) for *w*0/*W* = 0.005 and proximity factor, *G_p_*, according to *w*0*/W*.

**Figure 6 sensors-22-07761-f006:**
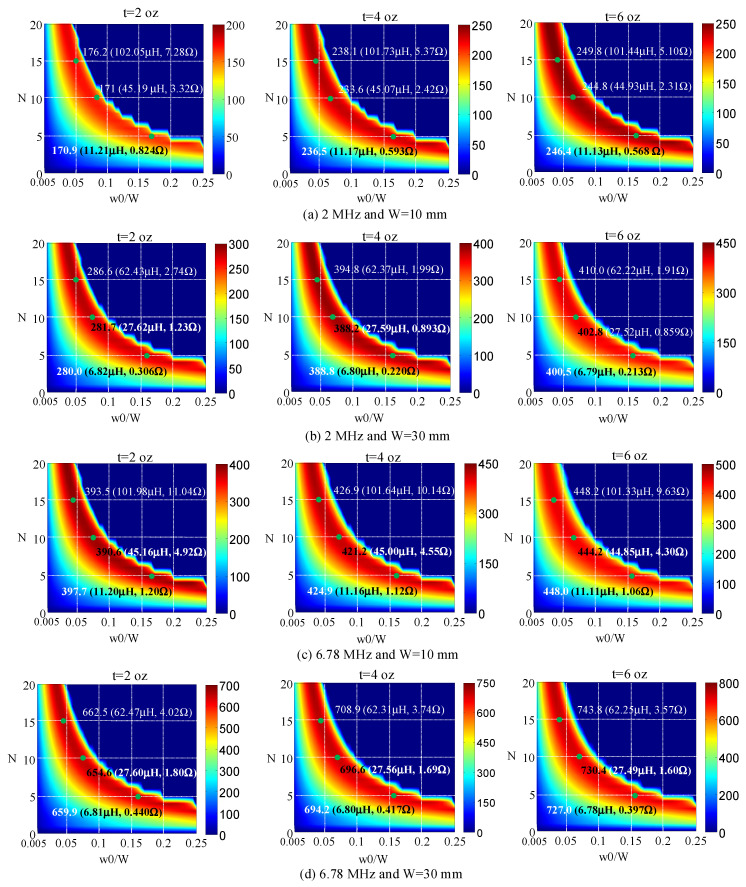
Calculated Q-factors according to 2 ≤ *N* ≤ 20 and 0.005 ≤ *w*0/*W* ≤ 0.25 for frequencies of 2 MHz and 6.78 MHz, W = 10 mm and 30 mm, and pattern thickness of 2 oz, 4 oz, and 6 oz.

**Figure 7 sensors-22-07761-f007:**
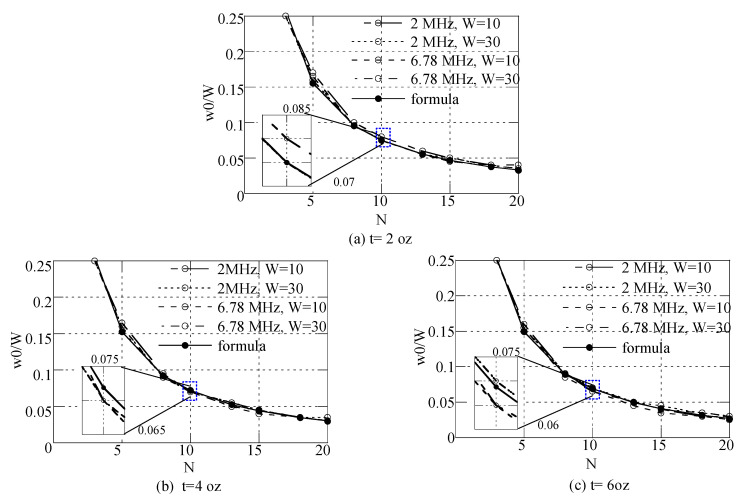
The formula’s optimal *w*0/*W*, calculated using (11), and comparison between the calculations and formula results according to turn number *N* for the maximum Q-factor at frequencies of 2 MHz and 6.78 MHz, total pattern widths (W) of 10 mm and 30 mm, and metal thicknesses (t) of (**a**) 2 oz, (**b**) 4 oz, and (**c**) 6 oz.

**Figure 8 sensors-22-07761-f008:**
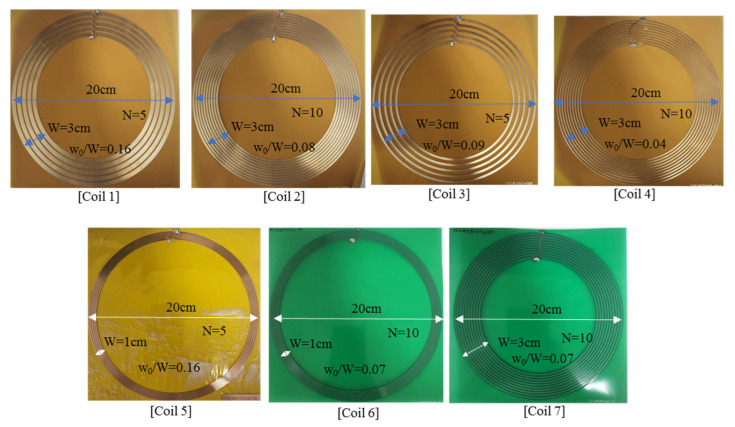
Photographs of coils optimized (coils 1, 2, 5, 6, and 7) for maximum Q-factor and fabricated coils that were not optimized (coils 3 and 4) to be compared with coils 1 and 2, respectively.

**Table 1 sensors-22-07761-t001:** Specifications of the fabricated coils, and comparison of the Q-factor at 2 MHz and ohmic losses of the coils between the measurement and calculation.

	t(oz)	N	*w*0/*W*	Calculation			Measurement		
				R (Ω)	L (μH)	Q-Factor	R (Ω)	L (μH)	Q-Factor
Coil 1	2	5	0.16 (W = 30)	0.3058	6.8153	280.0	0.344	6.92	252.77
Coil 2	2	10	0.08 (W = 30)	1.249	27.58	277.46	1.591	27.761	219.27
Coil 3	2	5	0.09 (W = 30)	0.426	6.98	205.69	0.584	7.354	158.24
Coil 4	2	10	0.04 (W = 30)	1.718	27.99	204.75	2.286	28.348	155.77
Coil 5	4	5	0.16 (W = 10)	0.595	11.17	236.15	0.751	11.275	188.6
Coil 6	4	10	0.07 (W = 10)	2.4247	45.07	233.6	3.067	45.34	185.73
Coil 7	4	10	0.07 (W = 30)	0.893	27.59	388.19	1.096	27.7	317.61

**Table 2 sensors-22-07761-t002:** Maximum coupling efficiency, mutual inductance, and *FoM* according to the distance between Tx and Rx coils for two cases. Case 1 is coil 5 (Tx) and coil 6 (Rx). Case 2 is coil 6 (Tx) and coil 6 (Rx).

Distance between Tx and Rx	Case 1 (Coil 5 (Tx) and Coil 6 (Rx))			Case 2 (Coil 2 (Tx) and Coil 2 (Rx))		
	M (μH)	FoM	Efficiency (%)	M (μH)	FoM	Efficiency (%)
15 cm	1.089	9.017	80.1	2.124	8.702	79.5
30 cm	0.217	1.797	34.5	0.475	1.946	37.2
45 cm	0.100	0.828	13.0	0.231	0.946	18.8

**Table 3 sensors-22-07761-t003:** Comparison between previous studies and this work.

	[14]	[15]	[16]	[17]	This Work
Verified frequency	Up to 1 MHz	Up to 1 Mhz	Up to 6.78 MHz	Up to 10 MHz	Up to 6.78 MHz
Calculated metal thickness	1 oz, 2 oz	1 oz, 2 oz	~4 oz	1 oz	2 oz, 4 oz, 6 oz
Coil shape	Circular	Circular	Rectangular	Rectangular	Circular
Optimal design formula	No	No	No	No	Yes

## Data Availability

Not applicable.

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
