# Peer review of "An Effective Design Formula for Single-Layer Printed Spiral Coils with the Maximum Quality Factor (Q-Factor) in the Megahertz Frequency Range"

_sensors, 2022, doi:10.3390/s22207761_

Round 1
Reviewer 1 Report
The technical objective and technical approach presented in this article appear quite similar to those in [11]. The authors are suggested to explicitly articulate and narrate this article's fundamental improvements (that is, not incremental improvements) with respect to [11]. At the moment, I recommend major revisions.
Author Response
Dear Reviewer:
We would like to thank you very much for your help in handling our paper. All your comments have been addressed and included in the revised version as detailed below. The changes are highlighted by red color text in the revision for your ease of comparison. And we hope the revision will be satisfactory.
Please see the attachment.

Reviewer 2 Report
The manuscript shows an optimal design formula to ensure the maximum Q-factor of printed spiral coil. Eqn.11 and Figure 7 in the manuscript are helpful for the related engineers for the design of magnetic resonant wireless power transfer applications, especially in MHz range.
Following problems should be addressed:
1. English should be improved. The authors seem to use the tense randomly.
2.Some equations should be derived in more detail, such as Eqn.2 to Eqn.5.
3. For the fitted Equation 7, the error should be analyzed.
4.Based on Table I, the results between the equation and the measurement are obviously different. Authors should analyze it in detail.
Author Response

(The authors gave the same response as above.)

Reviewer 3 Report
1.Manuscript is not contained much literature review.
2. The concept of this paper is very difficult to understand, so improve the quality of English presentation.
3. Comparison of results with existed work is not given.
4. More references are to be included.
Author Response

(The authors gave the same response as above.)

Reviewer 4 Report
The paper proposed a solution on maximizing the Q factor to improve the efficiency of WPT system. Some comments are listed as follows:
1. State the contribution at the end of your first section.
2. Briefly explain on your assumption of equation (6).
2. How the statement of line 118-119 reflect in real implementation?
3. Why the proposed equation only valid within the mentioned freq. (2MHz to 6.78MHz). What happen if the equation is applied to more than or less than the value. Explain on this.
4. High Q will definitely improve the efficiency. However, the obtained Q factor is very high (more than 100). I believe this will lead to a very narrow bandwidth. Hence, the proposed solution will be very sensitive to the parameter variations in the circuit. How do you recover the issue?
5. Do the proposed equation able to determine the most optimum value of Q factor?
Author Response

(The authors gave the same response as above.)

Round 2
Reviewer 1 Report
The authors have addressed my concern in their rebuttal and revised manuscript. I recommend this manuscript be published at Sensors.
Author Response
Thank you for your review and comments.
Reviewer 2 Report
All my concerns are addressed in the revised manuscript. It can be accepted.
Author Response
Thank you so much for your review and valuable comments!
Reviewer 4 Report
All the concerns have now been answered.
Minor concern is on the title. Maybe you may replace 'to ensure ' to 'for'. Your aim to obtain maximum Q or Optimum Q?
Author Response
Dear Reviewer,
We would like to thank you very much for your comments. Your comments have been addressed as follow.
Comment: Maybe you may replace 'to ensure ' to 'for'. Your aim to obtain maximum Q or Optimum Q?
Reply to the comment: Thank you for your comment.
The authors would like to change "to ensure" into "with".
The proposed paper aims to provide an effective design formula for single-layer printed spiral coils with the maximum Q-factor. The single-layer printed spiral coil which satisfies the formula has maximum Q-factor with total pattern width and metal thickness given.
Your favorable consideration would be very much appreciated.
Many thanks.